# Parental Attributions—Mothers’ Voices in Economically and Socially Disadvantaged Contexts

**DOI:** 10.3390/ijerph19159205

**Published:** 2022-07-27

**Authors:** Isabel Narciso, Sara Albuquerque, Maria Francisca Ribeiro, Luana Cunha Ferreira, Mariana Fernandes

**Affiliations:** 1CICPSI, Faculty of Psychology, University of Lisbon, Alameda da Universidade, 1649-013 Lisboa, Portugal; lcferreira@psicologia.ulisboa.pt; 2HEI-Lab, Lusófona University, 1749-024 Lisboa, Portugal; saramagalhaes9@msn.com; 3Center for Research in Neuropsychology and Cognitive and Behavioral Intervention, Faculty of Psychology and Educational Sciences, The University of Coimbra, 3004-531 Coimbra, Portugal; 4Ponto de Apoio à Vida, Clinical and Health Psychology, Faculty of Psychology, University of Lisbon, 1649-004 Lisboa, Portugal; mfranciscasrr@gmail.com; 5ProChild CoLAB, Campus de Azurém, 4804-533 Guimarães, Portugal; mariana.fernandes@prochildcolab.pt

**Keywords:** parental attributions, inadaptive parenting behaviors, children’s undesirable behaviors, parental reflexivity, socially and economically disadvantaged parenting

## Abstract

In the present study, the attributions of socially and economically disadvantaged mothers for their own negative parenting behavior and for their children’s undesirable behaviors as perceived by parents—understood as misbehavior—were analyzed. To this end, an exploratory study with a qualitative design was developed, in which 24 socially and economically disadvantaged mothers were individually interviewed. The data were analyzed following a thematic analysis approach, using software suited to qualitative analysis, namely NVIVO 12. The children’s undesirable behaviors as perceived by parents and some characteristics associated with parental performance (particularly the appraisal of the effectiveness of their negative practices) emerged respectively as external and internal factors, explaining mothers’ inadaptive behaviors—difficulties in behavior regulation, physical coercion, psychological control and paraverbal hostility. The parental subsystem and school emerged as the main external factors, and the psychological characteristics as the most relevant internal factors, explaining the children’s undesirable behaviors—challenge, immaturity, hostility, emotionally-based, school behavior/absenteeism and danger. The results also indicate weak self-critical reflexivity regarding some of the inadaptive behaviors. The comprehensive analysis of the results, based on the literature review, gave rise to an explanatory hypothesis on the dysfunctional circular process regarding the maintenance of inadaptive practices and children’s undesirable behaviors, considering the role played by parental attributions and by insufficient parental reflexivity.

## 1. Introduction

Socially and economically disadvantaged (SED) families are frequently exposed to multiple adversities (e.g., economic scarcity; poor housing conditions; drug and alcohol abuse; mental and physical health problems; exposure to family and community violence; social exclusion) [1,2]. Children who live in this context display poorer results in dimensions such as health (e.g., higher rates of childhood obesity and asthma), education (e.g., greater prevalence of school retention and dropout) and development (e.g., higher rates of teenage pregnancy) [1]. According to the Family Stress Model (FSM) [3,4], the relationship between SED contexts and child development is explained by the negative context of SED contexts in parents’ behavioral and emotional functioning (e.g., daily stress caused by financial constraints and psychological parental suffering), which consequently impacts interparental and marital relationships, as well as the quality of parenthood.

Several empirical studies have shown the vulnerability of families in SED contexts and the influence of the latter in the creation of negative parenting, characterized by rigid disciplinary practices with severe and physical punishment, criticism, and psychological control; poor affect expression; and little attention, responsivity, availability, support and quality time spent with the child [2,5]. Consequently, fewer secure attachments are promoted [1], thus negatively impacting the emotional security and development of children, who tend to present higher vulnerability to externalizing and internalizing problems and higher use of alcohol in adolescence [6,7].

Parental cognitions (e.g., beliefs, expectations, attributions) are also influential factors in parenting since they play a filtering role serving to restrain their behavior and regulate their relationship with their children [8,9]. Thus, as parental attributions act as filters that interpret and explain children’s behaviors, they subsequently influence parental responses [10].

The empirical literature points to a particular focus on the study of parental attributions for children’s behavior—rather than parental behaviors—placing more emphasis on the explanations for children’s negative rather than positive behaviors [11,12,13]. Negative attributions may focus on children, explaining their bad behavior through their personal characteristics or by emphasizing their negative intent (child-responsibility attributions);on parents, highlighting the parent’s role in the child’s bad behavior (parents’ causality attributions); or on both parents and children [10].

As far as children-focused attributions are concerned, the empirical literature has revealed an association between negative attributions for children’s undesirable behaviors as perceived by parents (CUBPP) (namely, as being intentional and caused by internal, stable and global factors) and negative and coercive parenting behaviors (e.g., hostile parental reactions) and less parental engagement in changing behaviors, which contributes to the perpetuation of their children’s behavioral problems [14,15,16]. On the other hand, the use of strategies with immediate recourse to punishment, instead of appeasing strategies for CUBPP, is associated with lower levels of trust, self-esteem and parental self-efficacy [12]. For example, studies with SED mothers have highlighted feelings of hopelessness towards children’s lack of response regarding their behavior, thus enhancing frustration and hostile behaviors [11].

SED parents attribute less responsibility to themselves for their children’s behaviors [10], in comparison to less economically disadvantaged parents. This situation might suggest less parental reflexivity, namely parents’ ability to critically reflect on parenthood, particularly on the exercise of parenting itself [8]. Thus, reflective functioning, i.e., the ability to understand and interpret one’s own and one’s children’s emotions [17,18,19] is also part of parental reflexivity [8]. Stern [20] found that low levels of reflexive functioning are associated with negative parental attributions for children’s behavior, reflected in their difficulty to control both their children’s and their own emotions. It should be noted that reflexive functioning is not common in SED contexts, most likely due to low levels of schooling and economic adversity [19]. The experience of chronic stress and adversity, as is common in these contexts, is associated with cognitive difficulties (e.g., memory and learning difficulties) [21], which encourages parents to automatically process the information [22] without considering the impact of the context on children’s behavior, thus perpetrating attributions of higher responsibility to the child [23].

Parents’ appraisal and integration of information concerning their children and subsequent response are also influenced by prior global (related to all children) and specific (uniquely related to their children) beliefs [23]. A considerable part of these beliefs and values are acquired in accordance with cultural standards (e.g., legitimacy of using force in relationships) and the family of origin (e.g., the importance attributed to physical punishment in children’s upbringing), and also considering their own childhood experience with harsh conditions (e.g., experiences of mistreatment) [9].

However, it should be noted that, even though parental cognitions are relevant, and particularly parental attributions, empirical studies with SED families are clearly insufficient [8]. The vulnerability of SED families to maladaptive parenting [5] and the role of parental attributions in parental responses [10] point to the importance of focusing on parental attribution in SED contexts. The present study, of an exploratory and qualitative nature, seeks to analyze SED mothers’ attributions in relation to their own negative parental behaviors and to their CUBPP.

## 2. Materials and Methods

### 2.1. Sample Collection Procedures

The sample was selected by means of convenience sampling and the collection process ended when the theoretical saturation point had been reached, i.e., the preliminary data analyses, concomitant with the data collection, revealed that the inclusion of new data would no longer add any relevant theoretical contributions [24]. The sample selection process involved the pre-establishment of the following inclusion criteria: (1) living in Portugal; (2) having children aged between 2 and 17 years; (3) living with their children or temporarily having children in foster care due to parental risk; (4) benefiting from social income due to their SED status.

The recruitment process was accomplished with the cooperation of several Portuguese social solidarity institutions, which invited mothers and fathers to participate in the present study after informing them of the purposes of the research and the interview conditions and procedures. All the interviews, which were individual and lasted an average of 90 min, were conducted by members of the research team and were held in the social solidarity institutions facilities. Prior to the interviews, the researchers supplied information on the conditions of the interview (procedures; the right to not have to respond to all the questions; audio recording), assured the participants of the confidentiality of their responses and requested their informed consent. Upon completion of the interview, the email address of the research team was provided so that the participants could contact the members to clarify potential doubts. Psychological support was also available through the Community Service of the Faculty of Psychology of the University of Lisbon (FPUL) Community Service. The present study was approved by the Ethical Committee of the FPUL.

### 2.2. Participants

This study sample consisted of 24 SED mothers (no fathers participated in the study). Some of the mothers had been referred to the Child and Youth Protection Services for parental risk (*n* = 8), while the remaining participants had not. The mothers’ age varied from 19 to 48 years (*M* = 32.54; *SD* = 8.05). Table 1 presents the participants’ most relevant sociodemographic characteristics.

As for the children living with their mothers (aged under 18 years)—17 girls and 31 boys)—eighteen mothers had one or two children (*n* = 8 and *n* = 10, respectively) and the remaining six mothers had three (*n* = 4) or four (*n* = 2) children. The average age of the children was 5.75 years (*SD* = 4.54). Two mothers reported having children with health problems and two mentioned developmental problems. As far as specialized technical support is concerned, five participants mentioned educational support in school, seven reported psychological support and three referred to speech therapy.

### 2.3. Instruments

Each participant responded to a semi-structured oral interview. At the beginning of the interview, a sociodemographic questionnaire was administered with the purpose of obtaining sociodemographic data (e.g., gender, age, parents’ and children’s academic qualifications; professional status, civil status, current relationship status; psychological support).

The interview script comprehended seven general topics from which questions were then developed: “good mother” meanings (e.g., “What, in your opinion, does it mean to be a good mother?”); parental self-characterization (e.g., “How would you describe yourself as a mother?”); perception of children (e.g., “I would like you to tell me about each of your children”); parental practices (e.g., “Here, on this piece of paper is a list of behaviors some parents have and some do not. I would like to talk to you about these behaviors: if this happens to you or not, when, why…”); sources of support (“What/who helps you to be a good mother?”); rewards, difficulties and challenges (e.g., “I would like you to talk about your main difficulties as a mother and how you deal with them”). Finally, the participants were asked to evaluate (justifying their answer) their parental satisfaction concerning their performance as mothers; their children’s behavior and their relationship with them (from 1: “very unsatisfied” to 5: “very satisfied”).

### 2.4. Data Analysis

The interviews were transcribed and imported using *NVIVO 12*, a software package used for qualitative data analysis. The data analysis mainly followed an inductive coding approach, which means the categories were defined based on the data collected and not on previous theories. A thematic analysis was conducted, following the procedures recommended by Braun and Clarke [25] and Maguire and Delahunt [26], as well as some general strategies that are characteristic of Grounded Theory, namely, the continuous comparative analysis process between data segments and interpretation, as well as the researcher’s reflexivity implemented by means of memos (i.e., written record of his/her thoughts; decisions and interpretations during the analysis process) enabling him/her to be attentive and aware of his/her influence in the research process [27,28]. The fact that the data coding involved three researchers—with continuous supervision by one of them, given the specialized knowledge in qualitative analysis—also contributed to controlling the influence of the researcher’s subjectivity.

In phase one, referred to as data acquaintance, the whole interview was read with the purpose of apprehending data holistically [25] and notes were taken on the ideas that emerged. The data coding phase [25] corresponded to the open coding [24,27,28] of segments or excerpts revealing parental attributions (either spontaneous or following questions that called for participants’ explanations of statements made about inappropriate parenting or unwanted child behavior). Thus, topics were identified, to which designations were attributed that coincided with words or phrases as close as possible to the meaning of each analyzed segment. As the analysis progressed, it was possible to discover similarities between the topics deemed relevant, which led to the emergence of concepts. In the search for themes phase [25], axial coding [24,27,28] was conducted, which implied intra- and inter-categorical analyses (concerning fisrt-order categories, meaning concepts). As for the larger scope categories,—second-order categories, or themes emerged—these included similar or different but related concepts. In phase four—thematic revision [25]—an in-depth comparative intra- and inter-thematic analysis (namely second-order intra- and inter-categories) was conducted to gear the analysis towards relevant patterns, which led to several reformulations of the initial categorical tree, thus leading to a fifth, more definitive phase in terms of emerging themes—defining and naming theme [25], allowing for a more detailed exploration of each particular theme. This more detailed exploration also allowed a second interactive and contextualized reading between some of the themes, corresponding to the initial phase of a possible process of selective coding leading to the development of theoretical explanations. Finally, the last phase—writing-up the data analysis [25]—resulted from the information emerging from the previous phases and was also accomplished with the inclusion of a literature review that was conducted during the research process.

## 3. Results

The results are organized based on two emerging themes: how do mothers explain their negative performance?; and how do mothers explain their children’s undesirable behaviors (CUBPP)? It should be noted that the categories are presented in italic and followed by brackets that indicate the number of participants coded, whenever equal to or greater than one-third of the participants (i.e., eight). Selected quotes illustrating categories are followed by: (1) one letter symbolizing the participant’s identification; (2) a number representing the age of the participant; and (3) the numbers that represent the age of the children. As an example, (L; 40; 12; 2; 1) indicates a quote from mother L, who is 40 years of age and has three children aged 12, 2 and 1 year(s), respectively.

### 3.1. How Do Mothers Explain Their Negative Performance?

#### 3.1.1. Type of Negative Parental Performance

Regarding the *type of negative performance* of mothers, a predominance of *difficulties in regulating children’s behavior* (10), and mostly *negative parental behaviors* (22) were observed. Difficulties in regulating children’s behavior (10) emerged as mainly being associated with situations of *permissiveness*, such as *absence of punishment*, *lack of firmness* and *excessive yielding to the children’s will*. Negative parenting behaviors were mainly associated with *hostile practices*, such as *physical coercion* (13) (e.g., ear pulling, smacking, slapping), *paraverbal hostility* (13) (e.g., yelling at the children) and *psychological control* (15), namely parenting behaviors that induce fear, shame and guilt (e.g., threats of abandonment, affective withdrawal, ignoring/ceasing to talk to the child, hostile scolding in public). Some mothers also reported behaviors such as *overprotection*, *unavailability,* and *instrumental affection* (i.e., fake expression of affection only manifested with the purpose of satisfying specific parental goals).

#### 3.1.2. Locus of Causality concerning Negative Parental Performance

The qualitative analysis revealed *external* and *internal attributions* for negative parental performance. As far as external attributions are concerned, the vast majority of the mothers attributed their negative performance to CUBPP (22)—“When she was naughty…like saying something weird I do not like” (Ai; 21; 3). However, the *well-being of children*, their *suffering*, their *dependence* and *developmental disorders* were also given as justifications. A minority of the mothers justified some of their negative parental behaviors on the basis of *cultural issues*, *learning acquired from their family of origin*, *financial difficulties* and *difficulties in couple relationship*.

Regarding internal attributions, almost all the mothers attributed their negative behavior to *factors related to parental performance* (21), namely *to the perception of the efficacy of practices* (11)—“So we regulate it that way and that is it, for me that is the basis. And I yell, I won’t say I don’t yell, I yell very much. «Oh, you are killing me, then when I die, you will go to an orphanage»…«Oh mother I don’t want that, I’m sorry»” (L; 40; 12; 2; 1); *personal characteristics associated with parenting* (11) (e.g., excessive empathy, lack of patience, self-control difficulties, being over-protective, etc.)—“I have never punished…as much as I may say I will do this or I will do that…my heart aches, I cannot punish…it’s a problem for me” (Ir; 28; 9; 4); *weariness* and *anger* (9)—“Because he is terrible! He’s at the stage where he thinks he can do what he likes (…) It makes me mad” (AS; 38; 15; 2); and *parental inefficacy of other practices* (9). Some participants also mentioned *disturbances in physical and psychological health*, their *life story* and *personal fatigue* resulting from their profession.

Concerning the relationship between the type of negative parental behavior and parental attributions, psychological control and physical coercion were mainly justified by CUBPP, by their efficacy and by the mothers’ weariness and anger. Paraverbal hostility was mainly attributed to its effectiveness, to CUBPP and also to personal characteristics associated with parenting. Internal attributions for difficulties in regulating the children’s behavior, which mainly included permissive behaviors, were predominantly marked by personal characteristics associated with parenting.

#### 3.1.3. Self-Critical Parental Reflexivity

Many of the mothers acknowledged the negativity of some of their practices, thus revealing *self-critical parental reflexivity* (15)—“One day I take his PlayStation away from him and the next I give it back and that’s not something you should do”–(T; 34; 13); “I realized that this [abandonment threat] is not a good thing to say because then they get psychologically frightened and think “she will abandon me because I did this or that”…” (Ma; 37; 14; 9; 5; 2). However, almost all the participants also demonstrated an *absence of self-critical parental reflection* (20) concerning diverse practices, without acknowledging their negativity—“I hit M. a lot. The little one only sometimes…but he deserves it!” (Tn; 35; 3; 1); “I have threatened, not that I would leave, but that he would go to his father” (Cl; 39; 10; 2).

Self-critical parental reflexivity of the mothers in the exercise of their parenting emerged associated with a recognition of the negativity of permissive and psychological control (mainly the threat of abandonment) parenting behaviors, paraverbal hostility and physical coercion. The absence of self-critical reflexivity was particularly associated with psychological control (mainly ignoring/ceasing to talk to the child), physical coercion and paraverbal hostility, given the perception of the effectiveness offered by such practices.

Considering the relationship between the locus of causality and parental reflection, only around half of the mothers who made external attributions were observed to have self-critical reflection in relation to some parental practices and, throughout their discourse, almost all of them revealed an absence of self-critical reflexivity, since they failed to acknowledge the negativity of several practices. As for the mothers who made internal attributions, even though the absence of self-critical reflexivity was predominant, it was possible to identify self-critical reflexivity in over half of the participants.

### 3.2. Attributions for Children’s Undesirable Behaviors as Perceived by Parents

#### 3.2.1. Type of Children’s Undesirable Behaviors as Perceived by Parents

Behaviors that are indicative of *defiance* were the most reported (19) (e.g., disobedience; complaining; stubbornness; tantrums)—“Once she was playing and asked me if she could go play with one of her classmates who lives behind us and I said “No L., we don’t know each other well yet” and she says to me “Damn it, you too!” (Ct; 30; 7; 4; 2). These behaviors were followed by other ones suggesting *immaturity* (12) (e.g., lack of responsibility and spoiled behaviors), hostile behaviors (11) (e.g., screaming, destroying objects and aggression among siblings), and behaviors indicative of *emotional difficulties* (9) (e.g., difficulties in emotional expression; relational avoidance, low self-esteem, low self-confidence, nervousness, sadness, insecurity)—“She is the most different from all of them, she is the quietest, sometimes we have to force her to talk to us, she doesn’t express herself as much as the other two; if she is upset sometimes she doesn’t say that she is upset (...)” (S; 21; 5; 3; 2).

Several *school-related* CUBPP were also mentioned by the participants (8), particularly bad school behavior and school absenteeism. A smaller number of mothers also mentioned *dangerous* behaviors. It should be noted that by referring to CUBPP, half of the participants (12) frequently added descriptions of characteristics or children’s positive behaviors, which was designated as *complementary appreciation of children*—“He is sometimes lazy (…) but he is a good kid” (AS; 38; 15; 2).

#### 3.2.2. Locus of Causality Relative to Children’s Undesirable Behaviors as Perceived by Parents

Almost all the mothers made *external* (20) and *internal* (21) *attributions* for their CUBPP. As regards *external attributions*, the *family* was highlighted as an explanatory factor by almost half of the mothers (13), in particular the *parental subsystem* (11), mainly parental performance (8), revealing self-critical parental reflexivity—“because I react badly maybe that encourages him to have that kind of behavior” (Cl; 39; 10; 2). A smaller number of mothers also referred to other causal factors concerning the parental subsystem, namely the *spouse’s parental performance*, the situation of *single parenting* related to the absence of a male parental figure, the *father of the child being in prison*, the *unavailability of the mother for professional reasons* and due to *inter-parental educational divergences*. Concerning the family system, some of the mothers also indicated the following as explanatory factors: the *fraternal subsystem*, highly associated with sibling interactivity; *the extended family*; the *conjugal subsystem*, namely the bad quality of the conjugal relationship and the separation of a spouse; and *circumstances of changes within the family*. In the *extra-familiar system* (13), *school* (11) (e.g., incompetency of teachers, relationship with peers, lack of school support) was the main explanatory factor for CUBPP.

As far as internal attributions are concerned, *psychological characteristics* (20), particularly *stubbornness/self-determination* (17)—“When we tell M. to pick something up, tell him he has to tidy up, and he says «you pick it up yourself » and he won’t do it (…).” (T; 35; 3; 1), emerged as a particularly important explanatory factor. Some mothers also mentioned other psychological traits, such as the *need for social acceptance*, *lack of effort/persistence*, *distraction*, *laziness*, *immaturity* and *perfectionism*. The *age* of children was also referred to as an explanatory factory by more than half of the participants (13) (R; 27; 10; 5; 3). Some of the mothers also highlighted other explanatory factors such as being *male*, having a *physical disease* and *psychological disorder*.

The mothers tended to use internal factors to explain the CUBPP related to danger, challenge and school—“Yes, even when he was little he was very adventurous, he is not afraid, he doesn’t think about the dangers.” (FP; 43; 12); “(…) «I am answering you (…) but you’re as stubborn as a mule», N. is stubborn” (L; 40; 12; 2; 1). However, it should be noted that the participants who made external attributions for defiant behaviors specifically mentioned the family subsystem as an explanatory factor, with particular emphasis on the parental subsystem—“J. is capable of being slightly more responsible than N., we always did everything for N., my mother and I even peeled his fruit” (L; 40; 12; 2; 1). Internal attributions for their CUBPP, particularly attributions related to psychological characteristics, emerged in relation to hostile practices (physical coercion, psychological control and paraverbal hostility)—“he doesn’t listen, so I have to scream, a lot!” (Ir; 28; 9; 4).

Hostility behaviors and those indicative of emotional difficulties were mainly justified by factors external to the children, particularly the family system, and specifically the parental subsystem. Some mothers also attributed children’s hostility to the conjugal subsystem (e.g., inter-parental conflicts) and to interactions within the sibling subsystem. The analysis of the mothers’ speech revealed that children’s hostility behaviors frequently emerged in relation to hostile parental practices.

The analysis of the mothers’ attributional speech revealed that they made either stable (19)—“no one can change the stubbornness of those two” (L; 40; 12; 2; 1), or unstable attributions (23)—“No, now that he is in school, it’s gotten worse” (J; 31; 7; 3) for CUBPP. The results indicated that the mothers make *stable and unstable attributions* for all types of CUBPP, although with some predominance of stable attributions—“for example, she has a tendency to raise her hand to me and then I have to scold her” (Ca; 26; 4; 2).

## 4. Discussion

The main purpose of the present study was to understand and characterize the attributions of SED mothers for their negative parental behaviors and CUBPP.

### 4.1. Attributional Maternal Windows with a View of Negative Parental Performance

The results showed that the participants justified their negative behaviors with both external and internal attributions. The fact that the external attributions focused predominantly on their CUBPP suggests cognitive unaccountability as far as negative parenting behaviors are concerned [29,30,31], thus fostering the maintenance of such practices. It should be noted that some mothers attributed their negative behaviors to cultural beliefs, learning acquired from their family of origin, referred to in the literature as parental cognitive schemes that condition the way parents perceive and evaluate their children’s behaviors and how they respond to them [9]. The attributions referring to financial difficulties and difficulties in the conjugal relationship are in line with the Family Stress Model [3,4], which highlights the impact of economic disadvantage on conjugal and inter-parental relationships, and consequently on parenting.

As for the internal attributions, mothers were found to justify their behaviors mainly by referring to factors related to their own parental performance, namely by an appraisal of the effectiveness of their practices, by personal characteristics related to parenthood and by their state of weariness and anger (even when related to their children’s negative behaviors). Attributions for disturbances in the mothers’ physical and mental health, their life story and professional work-related fatigue were also made, which corroborates the association of SED parenting with low levels of self-efficacy [32], psychological suffering [3,6], physical and mental health problems [33] and unfavorable personal experiences across the life course [23].

The findings referring to the type of negative parental behaviors, namely excessively permissive or hostile, are in line with the results of the empirical literature that point to a relationship between low income and social difficulties with the use of severe (characteristic of authoritarian styles) or permissive disciplinary practices [1,34]. The results of the present study show that the aforementioned practices are mainly related to the bad behavior parents wish to eradicate and are highly dominated by emotions, namely weariness and anger. The intense economic adversity experienced by the participants is, in itself, revealing of the exposure to high levels of stress [4], which may result in greater impatience and aggressiveness [6].

Moreover, low perception of parental self-efficacy, characteristic of disadvantaged contexts [32] associated with the perception of the efficacy of hostile practices primarily centered around here-and-now results (e.g., interrupting or eliminating CUBPP), may compromise reflective thinking and reinforce their maintenance [8,35]. The results revealed an inconsistent tendency toward parental reflexivity, since although half of the participants demonstrated some degree of self-critical reflexivity regarding some of the negative parenting behaviors in their discourse, almost all of them displayed an absence of reflexivity on several other negative behaviors. Some studies have shown that the experience of stress and adversity, as is common in economically disadvantaged contexts, contributes to a decrease in cognitive capacity [21,36], leading to the automatic processing of information that does not take contextual indicators into consideration [22], possibly leading to more impulsive and hostile parental behaviors. In this study, self-critical reflexivity was particularly low regarding the parental practices of psychological control, the impact of which may be less perceived by mothers, given the fact that it is not immediately visible when compared, for example, with physical coercion.

### 4.2. Attributional Maternal Windows with a View of Children’s Undesirable Behaviors as Perceived by Parents

The mothers’ explanations regarding their CUBPP focused on external factors—ranging from the family (with particular emphasis on the parental subsystem) and the extra-family context to internal factors—especially the psychological characteristics of the children.

The attribution of dangerous and defiant behaviors (as well as those related to school) for their own children (internal attributions) is in line with the empirical literature that has pointed to the predominance of negative perceptions of children in parents of children with behavioral problems which, in turn, are associated with disruptive practices that contribute to the maintenance of children’s negative behaviors [14,15,16]. In fact, in the present study, the analysis of the mothers’ discourse revealed a link between CUBPP of challenge and hostility and the use of hostile parenting practices. It should be noted that some types of negative cognitions promote negative emotional responses, especially when parents focus on symmetrical power dynamics in their relationship with their children [11], which may be explained by the overvaluing of obedience in disadvantaged contexts [30,37]. The exclusive attribution of CUBPP to children has also been found to lead parents to not reflect on their part in maintaining such behaviors or on other possible causes, thus reinforcing their negative perception of their children [38].

Even though the mothers made stable and unstable attributions for all types of CUBPP, the results show a slight predominance of stable attributions, which are particularly strong for challenging behaviors. The belief that such behaviors are permanent may increase parental stress, reinforcing negative attributions which, in turn, are related to more hostile parental practices [39]. It should also be noted that such attributions may result in parents’ low investment in changing their own behavior, since by making stable and internal attributions they punctuate the focus of change on their children and not on themselves [15], thus displaying weak parental reflexivity.

### 4.3. Crossed Attributional Windows—Reflexive Synthesis

A comprehensive interpretation of the results anchored on the literature review gave rise to an explanatory hypothesis on dysfunctional circular processes that contribute to the maintenance of CUBPP and negative parenting behaviors through the role played by parental attributions and the absence or insufficiency of parental reflexivity (Figure 1).

In relation to CUBPP, stable attributions for psychological characteristics, when associated with poor parental reflexivity (e.g., low self-critical sense; insufficient reflective functioning), lead to negative parental behaviors (e.g., physical coercion; psychological control) which, in turn, contribute to the maintenance of CUBPP which generate negative attributions in relation to their children, thus perpetuating this circular process. In turn, negative parental behaviors explained mainly by the bad behavior of children (external attributions), and/or by the positive parental appraisal of the effectiveness of their practices (internal attributions) associated with absent or weak parental reflexivity, reinforce such negative parental behaviors. The latter, as mentioned above, contributes to the maintenance of CUBPP, which lead to negative attributions in relation to the children, thus perpetuating the dysfunctional circularity.

It should be noted that, in the present study, this dysfunctional circularity process is particularly visible in relation to children’s challenging and hostile behaviors and does not generalize to other types of CUBPP, thus suggesting the absence of severe dysfunctionality in their parenting. The results indicating that the mothers in the present study do not have severe deficits in their reflective capacity are also worthy of note. In fact, some degree of self-critical reflexivity on the part of the mothers was observed, especially regarding physical coercion practices as well as the negative contribution of the parenting subsystem to the children’s emotional difficulties.

### 4.4. Limitations and Strengths

Some limitations should be taken into consideration. As this is a qualitative study, the data collection, analysis, and discussion processes were exposed to the subjectivity of the researchers involved. Attempts were made to overcome this limitation by the constant self-scrutiny of the principal investigator through the use of memos, i.e., written records of their reflection process and decision making [24,27,28], but also by the involvement of three researchers in the data analysis, one a senior specialist in qualitative methodologies.

With regard to the sample, the fact that it is a convenience sample carries some limitations. Although the number of participants is not insufficient, as the theoretical saturation point was reached [24], a larger sample and the inclusion of mothers and fathers would have enriched the study by including different subgroups that would have enabled a comparison of results regarding several important variables: reasons for signaling (due to economic issues vs. risky parenting), parents and children’s gender, and children’s age. Finally, the mothers’ participation was voluntary, which may imply that mothers with serious parenting problems did not participate in the study due to their potential discomfort in discussing their experiences.

Despite the aforementioned methodological limitations, this study contributes to the development of the area of parental attributions, increasing the knowledge of the description and impacts of parents’ attributions for their own negative parenting behaviors, a particularly scarce feature in the current literature. In addition, it promotes reflection on the complexity of this area by raising the possibility of dysfunctional circular processes that contribute to the maintenance of CUBPP and negative parenting behaviors. Additionally, the focus on the attributions of parents in SED situations is particularly relevant given the particular risk for negative parenting in this context [5]. Furthermore, the emphasis on the role of parental reflection on both their own and their children’s behaviors, as well as on the associated attributions, is one of the main innovative features of this study. Finally, as a qualitative study, a deeper and more reliable understanding of these processes is offered, given that it focuses on the description of parents’ experiences and perceptions, thus reducing the risk of bias inherent to the completion of self-report questionnaires, particularly in low literacy populations (as is the case in SED contexts).

### 4.5. Implications for Research and Intervention

Regarding future studies using this same sample, it would be of interest to analyze the attributions for children’s positive and mothers’ adaptive behaviors, since, to our knowledge, there is an absence of empirical studies on this topic. It would also be pertinent to expand studies on parental reflexivity, particularly its influence on processes of change in negative parental behaviors. Finally, it would also be important to conduct longitudinal research in order to study in depth the hypothesis emerging from this study regarding the dysfunctional circular processes that contribute to the maintenance of CUBPP and negative parenting behaviors through the role played by parental attributions and the absence or insufficiency of parental reflexivity. Additionally, in future studies, mediators of mother–child interactions should be included. For example, father’s involvement could be a variable of interest, given the assumption that mother–child interaction and father–child interaction within the same family are linked [40]. This is also in line with the family system theory model [41] or the spillover model [42], which assert that the behavior in one family relational setting (e.g., father–child) can also manifest in other family relationships (e.g., mother–child).

Considering the findings of the present study and the conclusions of Butcher and Niec [43] that changing parental attributions in specific situations can alter general attribution patterns for children’s behaviors, interventions targeting attributions appear to be of utmost importance. The use of narrative strategy frameworks may be relevant, since in order to generate change, it is essential to alter dominant parental narratives regarding children’s behaviors and their own conditioning behaviors [44]. Moreover, such strategic frameworks include a more analogical language—stories, metaphors, proverbs, images, photographs, games, movies, simulations, etc.—which is appropriate for people with low literacy levels. The use of video recordings of parent–child interactions targeting specific intervention on dysfunctional attributional patterns is also appropriate [43].

## 5. Conclusions

The present study highlights the role of attributions and reflexive parenting, not only in relation to CUBPP, but also in relation to maladaptive parenting practice itself. Overall, severe dysfunctionality in parenting and severe deficits in reflexive capacity are not found in SED mothers. However, the manifestation of dysfunctional circularity processes was highlighted particularly in relation to the children’s challenging and hostile behaviors. The attribution of the latter for the psychological characteristics of stable children, associated with the perception of the self-efficacy of their inadaptive parenting practices, without reflexivity on their emotional impacts on children, reinforces the perpetuation of cycles of inadaptive parenting practices and CUBPP and their attributions. Thus, it is essential to focus not only on the description of maladaptive parenting practices and CUBPP but, above all, on the dominant parental attributions in these two domains and the circularity involved in them.

## Figures and Tables

**Figure 1 ijerph-19-09205-f001:**
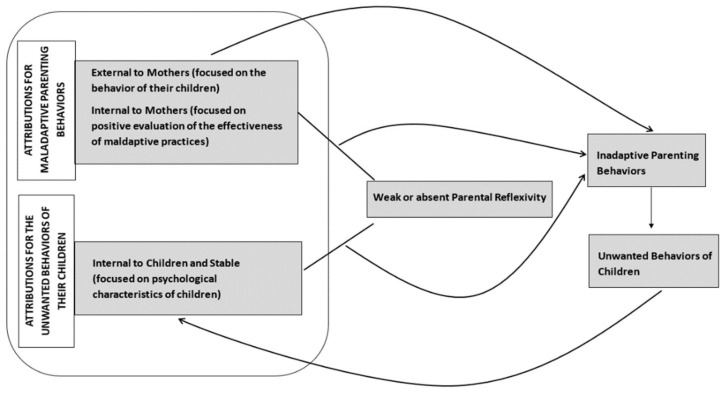
Explanatory hypothesis on dysfunctional circular processes.

**Table 1 ijerph-19-09205-t001:** Sample’s sociodemographic characteristics.

Characteristic (*n*)		
Ethnicity	Relational Status	Psychological/Psychiatric Support:
Caucasian (13)	Married/Cohabitation (17)	Never (22)
African (8)	Divorced/Separated (5)	Current or previous (2)
Roma (3)	No marital relationship (2)	
**Professional Status**	**Years Spent in Education**	**Mother’s Health Problems**
Employed (5)	<5 years (2)	No (22)
Unemployed (17)	5–6 years (6)	Yes (2)
Retired (1)	7–9 years (7)	**Religiosity**
Student (1)	10–12 years (9)	Believer (18)
	Higher Education (0)	Non-believer (6)

## Data Availability

Not applicable.

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
