# Peer review of "Parental Attributions—Mothers’ Voices in Economically and Socially Disadvantaged Contexts"

_ijerph, 2022, doi:10.3390/ijerph19159205_

Round 1

Reviewer 1 Report

This is a sound little study that is very much worth publication.   For note is the following:

1. the phrase 'the impact of the latter on negative parenting' p.2, line 51, ought to read 'the influence of the latter in the creation of negative parenting', no?  Otherwise, it could be read that 'negative parenting'  existed before and outside of socio-economic impacts and influences.

2. The brief mentions of the gendered nature of the sample do not see quite enough.  Firstly, I was struck by the many more references to mother and son conflicts than those of mother and daughter.  A couple of lines that give attention to this would be very useful.  Secondly, though the absence of fathers is noted, again, I believe that this too needs a couple of lines given the importance of postively invovled fathers in mitigating mother-child conflict.  These additions should come with the relevant citations.

Otherwise, well done.     

Author Response

This is a sound little study that is very much worth publication. For note is the following:

1.the phrase 'the impact of the latter on negative parenting' p.2, line 51, ought to read 'the influence of the latter in the creation of negative parenting', no?  Otherwise, it could be read that 'negative parenting'  existed before and outside of socio-economic impacts and influences.

It was altered as suggested.

  1. The brief mentions of the gendered nature of the sample do not seem quite enough.  Firstly, I was struck by the many more references to mother and son conflicts than those of mother and daughter. 

We agree with the reviewer’s comment and therefore more evidence of mother-daughter interactions (quotations from participants) was included.

Secondly, though the absence of fathers is noted, again, I believe that this too needs a couple of lines given the importance of positively involved fathers in mitigating mother-child conflict.  These additions should come with the relevant citations.

This suggestion was included as directions for future studies in the discussion section.

Otherwise, well done.

Thank you.

Reviewer 2 Report

Dear Authors,

the topic of your research is very interesting. The structure of the content is logical. The content fits to the title of the paper. The literature review introduces the reader into the topic. The chosen methodology meets the research objectives. In my opinion, you conducted too many interviews for qualitative research, especially for one paper. I assume that the number of respondents chosen was justified by the saturation of the data. In terms of sampling, I would have some additional questions. However, they would not affect my assessment of the paper. The way in which qualitative data is compiled is methodologically correct.

The discussion and the conclusions are also very well developed. I read the paper with great interest. I recommend the paper to be published in the given form.

Kind regards,

Reviewer 

Author Response

Thank you for your comments. We would just like to confirm that the number of participants was justified by the saturation of the data.

Reviewer 3 Report

At first glance, a standard scientific article in the given scientific field, based on qualitative research, but with interesting results, achieved through a relatively sophisticated in-depth analysis, scientifically relevant. On the substantive side, I have no comments, in my opinion the authors proceeded exactly, they could only have described the research model in more detail, in the sense of the qualitative research method in greater detail. I agree with the argumentation and presented results, I appreciate the discourse reflected in the Discussion. The article as a whole is interesting and worthy of publication.

Author Response

Thank you for your comments. As suggested, we have described the research model in more detail, particularly the data analysis of the qualitative research method.

Reviewer 4 Report

Dear authors, many thanks for a comprehensive paper. I only have minor comments to make.

- I did not fully understand the inclusion criteria “living with their children or having children in temporary foster care due to parental risk”. It seems to imply some comparison. Or is the emphasis meant to be on “temporary” foster care in the second instance, i.e. that if they are not living with their children it is merely a temporary situation?

- In Table 1, perhaps it might be easier to label “academic qualifications” as “years spent in education” or similar?

- “Health problems” in Table 1 – does this refer to health of mothers (which I first thought) or health of children (as per the following paragraph)?

- “the categories are presented followed by brackets that indicate the number of participants coded, whenever equal to or greater than one third of the participants (i.e., eight).” However, in the first results paragraph you give a 7 in brackets. This seems to contradict the bracket rule? Or is this an error?

Author Response

- I did not fully understand the inclusion criteria “living with their children or having children in temporary foster care due to parental risk”. It seems to imply some comparison. Or is the emphasis meant to be on “temporary” foster care in the second instance, i.e. that if they are not living with their children it is merely a temporary situation?

Exactly, if they are not living with their children, it is merely a temporary situation. We change the phrase to: “temporarily having children in foster care due to parental risk”.

- In Table 1, perhaps it might be easier to label “academic qualifications” as “years spent in education” or similar?

It was altered, as suggested.

- “Health problems” in Table 1 – does this refer to health of mothers (which I first thought) or health of children (as per the following paragraph)?

It refers to mother’s Health Problems and the phrase was altered accordingly.

- “the categories are presented followed by brackets that indicate the number of participants coded, whenever equal to or greater than one third of the participants (i.e., eight).” However, in the first results paragraph you give a 7 in brackets. This seems to contradict the bracket rule? Or is this an error?

It was an error. We thank you for pointing that out.
